# Using fluorescence flow cytometry data for single-cell gene expression analysis in bacteria

Luca Galbusera[1,2], Gwendoline Bellement-Theroue[1,2], Arantxa Urchueguia[1,2], Thomas Julou[1,2]*, Erik van Nimwegen[1,2]*

**1** Biozentrum, University of Basel, Basel, Switzerland, **2** Swiss Institute of Bioinformatics, Lausanne, Switzerland

* thomas.julou@unibas.ch (TJ); erik.vannimwegen@unibas.ch (EVN)

**Data Availability Statement:** No primary data were created for this study, and we refer to the articles from which the data used in our manuscript stem.

## Abstract

Fluorescence flow cytometry is increasingly being used to quantify single-cell expression distributions in bacteria in high-throughput. However, there has been no systematic investigation into the best practices for quantitative analysis of such data, what systematic biases exist, and what accuracy and sensitivity can be obtained. We investigate these issues by measuring the same *E. coli* strains carrying fluorescent reporters using both flow cytometry and microscopic setups and systematically comparing the resulting single-cell expression distributions. Using these results, we develop methods for rigorous quantitative inference of single-cell expression distributions from fluorescence flow cytometry data. First, we present a Bayesian mixture model to separate debris from viable cells using all scattering signals. Second, we show that cytometry measurements of fluorescence are substantially affected by autofluorescence and shot noise, which can be mistaken for intrinsic noise in gene expression, and present methods to correct for these using calibration measurements. Finally, we show that because forward- and side-scatter signals scale non-linearly with cell size, and are also affected by a substantial shot noise component that cannot be easily calibrated unless independent measurements of cell size are available, it is not possible to accurately estimate the variability in the sizes of individual cells using flow cytometry measurements alone. To aid other researchers with quantitative analysis of flow cytometry expression data in bacteria, we distribute *E-Flow*, an open-source R package that implements our methods for filtering debris and for estimating true biological expression means and variances from the fluorescence signal. The package is available at https://github.com/vanNimwegenLab/E-Flow.

## Introduction

It is has become well recognized that, due to the intrinsic stochasticity of the gene expression process, even isogenic populations of microbial cells growing in homogeneous environments exhibit significant heterogeneity in their gene expression, e.g. [1–4]. Therefore, the traditional

Therefore, all relevant data are within the manuscript.

**Funding:** This work was supported by the Swiss National Science Foundation in the form of a grant awarded to EvN (31003A 159673). Calculations were performed at sciCORE (http://scicore.unibas.ch/) scientific computing core facility of the University of Basel, and flow cytometry was performed at the FACS core facility of the Biozentrum. The funders had no role in study design, data collection and analysis, decision to publish, or preparation of the manuscript.

**Competing interests:** The authors have declared that no competing interests exist.

studies at the population level, by smoothing out this heterogeneity, tend to hide crucial information [5, 6] that is required to correctly understand and interpret the observed behavior of microbes [7].

Although most studies of single-cell gene expression in bacteria use fluorescent reporters in combination with microscopy to quantify gene expression in single cells, fluorescence flow cytometry (FCM) is also an attractive alternative methodology for single-cell gene expression studies in bacteria. In particular, given that flow cytometers can quantify the fluorescence of thousands of cells per second, flow cytometry allows for high-throughput characterization of the single-cell expression distributions of a large number of fluorescent reporters [8, 9]. Indeed, in recent years there has been a large number of studies in which standard commercially available flow cytometers were used in combination with fluorescent reporters to measure gene expression at the single-cell level in bacteria [10–31], as well as in single-celled eukaryotes [32, 33].

However, so far there has been little systematic investigation into the accuracy of flow cytometry in quantifying gene expression in single cells, or a systematic comparison with the results from microscopy measurements. Here we aim at filling this gap by systematically comparing flow cytometry measurements with measurements from a microscopy setup. In particular, there are several technical challenges in analyzing fluorescence flow cytometry data of individual bacterial cells:

1. *Differentiating cells from debris.* Bacterial cells are typically one thousandth the volume of mammalian cells, which places them near the edge of instrument detection. At this size it can be challenging to differentiate viable cells from debris of similar size [9, 34–37]. In the literature different approaches are used to separate debris from viable cells. Most of these approaches use *ad hoc* combinations of the scatter measurements to retain a fraction of the measurements.
   We here perform a careful analysis of all the scatter signals reported by the flow cytometer and propose a principled way of identifying debris from viable cells using a Bayesian mixture model that considers all the information available in the scatter signals.

2. *Distinguishing measurement noise from biological variability.* In order to quantify the amount of biological gene expression variation in a population of isogenic cells, it is important to quantify to what extent variation in measured fluorescence intensity derives from biological variation, and to what extent it derives from measurement noise.
   We show that flow cytometry measurements contain a substantial amount of shot-noise which can be easily mistaken for true biological variability, and develop a method to correct for this shot-noise using measurements of reference beads that are commonly used to calibrate flow cytometers. Using a mixture modeling approach, we develop a rigorous method for estimating the true mean and variance in expression levels of a population of cells.

3. *Accounting for autofluorescence.* Because most genes are expressed at low levels in bacteria (roughly one per cell cycle or less for half of the genome [38]), the relative fluorescence produced by fluorescent proteins compared to autofluorescent compounds is very low for many reporters [39]. Therefore, gene expression estimates require careful correction for autofluorescence.
   We here provide methods for correcting both the estimated mean and variance in fluorescence levels for autofluorescence using measurements of cells that do not express GFP.

4. *Estimating the distribution of GFP concentrations.* While we provide methods for accurately estimating the distribution of total GFP levels in a population of cells from the flow cytometry measurements, microscopy measurements show that total GFP levels correlate strongly

with cell size and that GFP concentrations vary significantly less across cells than total GFP. Estimating the distribution of GFP concentrations directly using flow cytometry requires to not only estimate the total GFP but also the volume of individual cells. Although forward- and side-scatter signals can be used to distinguish the average size of populations of cells of sufficiently different shapes and sizes [40–43], it is substantially more challenging to accurately quantify the relatively small cell-to-cell variations in cell volume for populations of isogenic bacteria growing in a homogeneous environment. In line with previous works [35, 44–46] we find that, because forward- and side-scattering measurements depend on cell volume in a complex non-linear manner and contain a substantial amount of shot noise that cannot be easily calibrated, it is impossible to accurately quantify the sizes of individual cells. Consequently, it is not possible to directly estimate the distribution of GFP concentrations from flow cytometry measurements. However, we show that because GFP concentrations and cell sizes fluctuate approximately independently, it is still possible to obtain reasonably accurate quantifications of the *relative* amounts of GFP concentration fluctuations for different genes.

Although the precise flow cytometer used will of course affect the precise values of the measurements and calibrations, the methods for separating true cells from debris, estimating and correcting for autofluorescence, and correcting for measurement shot noise, are general and should be applicable to data from most flow cytometers. Our methods have been implemented as an R package called *E-Flow*, which we make publically available and can be easily integrated in any flow cytometry data analysis pipeline.

## Materials and methods

### Strains and growth conditions

We measured the fluorescence distributions for a number of different *Escherichia coli* MG1655 strains carrying fluorescent transcriptional reporters (a GFP gene downstream of a given promoter, either on a low-copy number plasmid, or integrated into the chromosome) both using flow cytometry of batch cultures and time lapse microscopy in a microfluidic device (Mother Machine). We considered a number of different promoters, that have different means and variances of expression levels.

In particular, we considered *E. coli* strains with a lacZ-GFP fusion integrated in the chromosome [47], and a set of *E. coli* strains that carry a transcriptional reporter expressed from a low copy number plasmid [48]. These reporters included known target promoters of the LexA transcription factor (dinB, ftsK, lexA, polB, recA, ruvA, or uvrD) [49] and two synthetic promoters that were obtained by experimental evolution and express at levels corresponding to the median and the 97<sup>th</sup> percentile of all native *E. coli* promoters [23]. Throughout the paper, we refer to these two synthetic promoters as high and medium expressers.

To estimate autofluorescence in both the FCM and microfluidic experiments, we used two strains that carry plasmids where the GFP sequence is downstream of a random sequence (pUA66 and pUA139) [48] and hence do not express GFP [23].

In the microfluidic experiments, cells carrying a lacZ-GFP fusion were tracked using time-lapse microscopy while growing in a microfluidic device in M9 minimal media supplemented with 0.2% lactose (which leads to full induction of the lac operon), taking measurements every 3 minutes [47]. Detailed experimental procedures are available in the corresponding publication [47]. Microfluidic experiments with strains carrying a transcriptional reporter expressed from a plasmid were performed following the same procedure, using M9 + 0.4% glucose

(supplemented with $50\mu g$ / mL of kanamycin during the overnight preculture only) and acquiring data over 4 hours.

To obtain comparable measurements with flow cytometry (FCM), the same strains were grown in the same conditions as for the microfluidic measurements. Practically, strains expressing from a plasmid were inoculated from frozen glycerol stocks and grown overnight in $200\mu L$ of M9 + 0.4% glucose supplemented with $50\mu g$/mL of kanamycin. After 100× dilution in fresh medium without kanamycin, strains were grown to saturation again, and re-diluted 100× to fresh medium without kanamycin. For the lacZ-GFP strain, we used $200\mu L$ of M9 + 0.2% lactose with only one overnight culture. For all strains, expression was measured in mid-exponential phase (typically after 4h), adjusting the cell concentration with PBS if necessary. All cultures used for FCM measurements were incubated in 96-well plates at 37˚C with shaking at 600-650 rpm.

To study the accuracy of the scatter signal for estimating cell size, we used the data acquired for a previous project in the lab [31] where both flow cytometry measurements and microscopy measurements of cell size distributions have been obtained in four different media characterized by different size distributions: M9 supplemented with either 0.2% glucose (w/v), 0.2% glycerol (v/v) or 0.2% lactose (w/v); a MOPS based synthetic rich media (Teknova, M2105) supplemented with 0.2% glucose. We refer to the original study for more information about the cell cultures and growth conditions [31].

## Flow cytometry

The flow cytometry measurements were obtained with a BD FACSCanto II cytometer and were managed using the Diva 8 software. The excitation beam for the GFP was set at 488 nm and the emission signal was captured with a 530/30 nm bandpass filter. The gain voltage were set by default to 625V, 420V, and 600V for FSC, SSC, and GFP acquisition respectively, and events were created for measurements where $FSC > 200$ & $SSC > 200$. For each sample, $5 \times 10^4$ events were recorded at a typical flow rate ranging from $1 \times 10^4$ to $2 \times 10^4$ per second.

## Calibration beads

CS&T (Cytometer Setup and Tracking Beads) are artificial fluorescent beads that are used to calibrate fluorescence measurement values [50]. To calibrate the measurement shot noise we used beads of lot 41720 that contains beads of two different sizes, which have high, medium ($3\mu m$ in size) and low fluorescence ($2\mu m$ in size) levels.

## Microscopy size estimation

To estimate cell sizes, strains containing a plasmid without promoter were selected from 4 different media with different size distributions (M9 + glucose, lactose or glycerol; MOPS + glucose. See *Strains and growth conditions*). Cells were then placed on a 1% agarose pad and phase contrast images were obtained with a Nikon Ti-E microscope using a $100 \times$ Ph3 objective (NA 1.45) and an Hamamatsu Orca-Flash 4.0 v2 camera. Cell outlines were identified using a custom MATLAB pipeline [31].

## R package *E-Flow*

The analysis pipeline presented in this paper has been implemented in the R package *E-Flow* available on GitHub https://github.com/vanNimwegenLab/E-Flow. Here the methods were tested with flow cytometers manufactured by BD and operated through the DIVA software.

Nonetheless we kept the methods as general as possible, such that they should be applicable to flow cytometers of other manufacturers.

For a detailed explanation of the package, we refer to the GitHub page, including the vignette and the documentation of the individual functions. Here we list the main components of the software:

1. *Filtering*: The cells are filtered based on their scattering profile and an estimate of the mean and variance of the population is obtained. This is the most resource-intensive step and therefore can be parallelized.

2. *Mean and variance*: The mean and variance of the population of cells is computed. Measurements that are outliers in the fluorescence are accounted for using a mixture model.

3. *Autofluorescence removal*: Using the fluorescence distribution of non-expressing cells, an estimate of the autofluorescence is obtained and subtracted from the mean and variance of the population.

4. *Shot noise removal*: The shot noise introduced by the machine is removed and a corrected variance is calculated. This can be regarded as a proxy for the biological gene expression noise.

## Results

### Signals reported by the cytometer

In flow cytometry, a beam of light is used to illuminate cells that flow one by one through a channel; a series of detectors is able to record the light scattered by the single cells at right angles or in the forward direction and the cell fluorescence stimulated by the incident light beam. Most flow cytometers, including the BD Canto II used here, report for each measured 'event' (typically corresponding to a single measured cell) a forward-scatter signal, a side-scatter signal, and a fluorescence signal. Each of these signals is in turn represented by 3 statistics of the electrical impulse, namely height, area, and width of the impulse (Fig 1). The height corresponds to the maximal value of the impulse, the area to the area under the curve and the width is its time duration [51] (see Section 1.1 in S1 File).

We noticed that these statistics are not all independent. In particular, for all three signals, the area is always directly proportional to the product of height and width (S1 Fig and Section 1.2 in S1 File). Moreover, while height and width vary approximately independently across events, the area correlates significantly with both (S2 Fig in S1 File). Therefore, we only use height and width for the subsequent analysis of the forward- and side-scatter signals.

For the fluorescence signal we were unable to find any systematic dependence between the width of the fluorescence signal and any biological signal, such as cell size or total fluorescence. In addition, for the calibration beads there is clearly no information in the width of the fluorescence signal (S3 Fig and Section 1.3 in S1 File). Therefore, for the fluorescence signal we will only use the height statistic as a proxy for the total fluorescence of the cells. While we believe that all these considerations apply generally to flow cytometers, we also observed anomalous behavior of the signal at very low fluorescence levels that may be specific to the BD machine used here (see Section 1.4 in S1 File). Due to this anomalous behavior, quantitative analysis is restricted to constructs for which the GFP fluorescence is at least as high as the autofluorescence of the cells (see S4 Fig in S1 File).

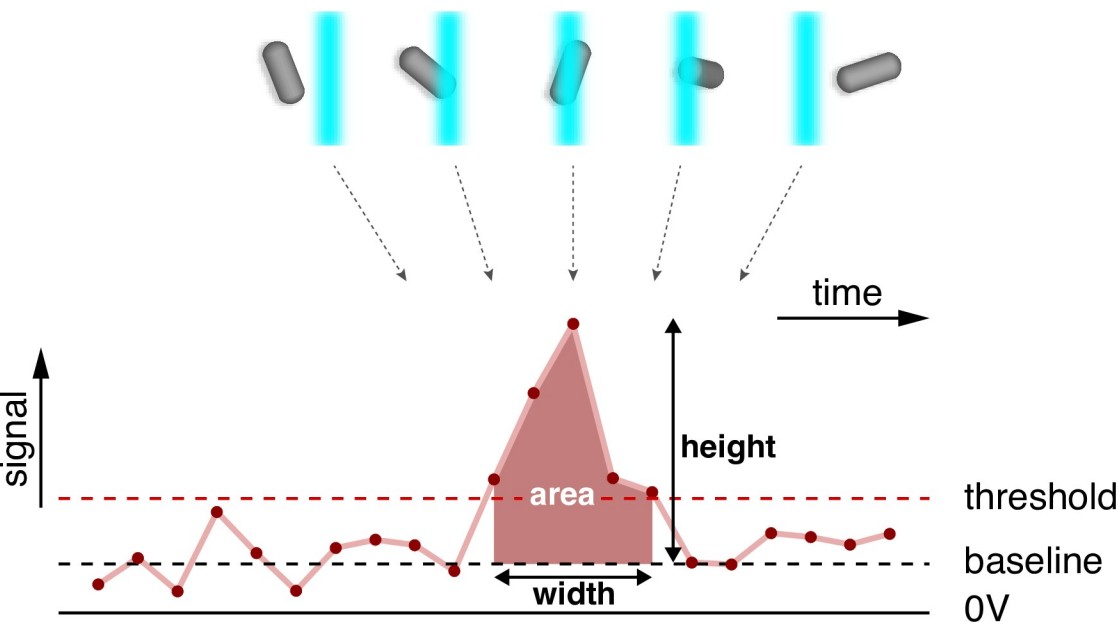

**Fig 1. The signals reported by the cytometer.** As a particle enters the laser beam, an electric signal (pulse) is generated which reaches its maximum when the particle is in the middle of the beam and trails off as the particle leaves the beam. Each pulse with height over a certain threshold is recorded and three quantities are reported: height, area, and width of the pulse.

## Filtering events based on their forward- and side-scatter

In comparison to eukaryotic cells, bacterial cell produce only relatively weak scattering signals, and we used permissive settings of the device to call events. This increases the likelihood of having spurious observations that correspond to non-viable cells and other debris. Consequently, we needed a strategy for using the measured forward- and side-scatter of the events to separate viable cell measurements from debris. As explained above, the scatter of each event is characterized by 4 statistics, namely the height and width of both the forward- and side-scatter. Thus, the measured scatter of each event can be represented by a point in a 4-dimensional space, and a given dataset corresponds to a distribution of points in this 4-dimensional space. To separate viable cells from debris we fit this distribution with a mixture of a multivariate Gaussian distribution and a uniform distribution, as detailed in the Section 2 in S1 File. The rationale behind this mixture modeling is that most of the data represents good cells and should cluster in this 4-dimensional space, whereas the outliers are relatively rare and more widely distributed. In this model, the Gaussian part of the mixture captures the cluster of good cells, while the uniform component takes care of outliers, i.e. fragments of dead cells and other debris.

Fig 2 shows 2D projections of the 4D scatter of forward- and side-scatter for events taken from *E. coli* cells that carry a lacZ-GFP fusion (see [47] for a description of the strain used) while growing in M9 minimal media supplemented with lactose. Besides the scatter of measurements, Fig 2 also shows the multivariate Gaussian fitted to the data, showing that this Gaussian indeed captures the bulk of the measured events.

Once the mixture model has been fitted to a dataset, a posterior probability $p_i$ is calculated for each measured event $i$ to correspond to a viable cell, i.e. the probability that the observation derives from the multivariate Gaussian component of the mixture as opposed to deriving from the uniform distribution. By default the *E-flow* software retains all events with posterior

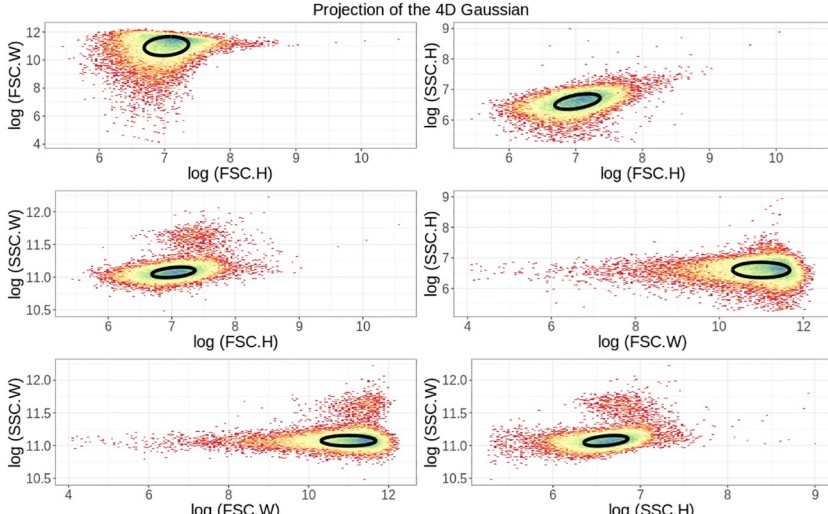

**Fig 2. Mixture model fitting of the scatter signals.** The panels show different two-dimensional projections of the full 4D distribution of heights (H) and widths (W) of forward- (FSC) and side-scatter (SSC) measurements for $5 \times 10^4$ events obtained from *E. coli* cells growing in M9 minimal media with lactose. The ellipses show the contour of the fitted multivariate Gaussian distribution, one standard deviation away in each principal direction. Note that the color indicates the local density of points.

probability $p_i \geq 0.5$ and discards as outliers events with $p_i < 0.5$, but the user can change this threshold probability if desired. S5 Fig in S1 File shows the same scatter of measured events as shown in Fig 2, but now with selected events in red and events that were filtered out in black when using the default threshold of $p = 0.5$.

As the forward- and side-scatter should reflect the size, shape and composition of the objects measured in each event, one may wonder to what extent filtering out events based on their forward- and side-scatter may bias measurements towards cells of a certain size. Indeed, in previous work, e.g. [19], researchers have attempted to select subsets of cells with similar shapes and size by very strictly gating on forward- and side-scatter, retaining only those cells that lie near the center of the Gaussian distribution. To check the viability of such an approach, we compared the distribution of measured fluorescence levels with two extreme filtering strategies: one very lenient in which all events with $p > e^{-10}$ are retained and one very strict in which only cells with $p > 1 - e^{-10}$ are retained. As shown in S6 Fig in S1 File, there is virtually no difference in the observed distribution of fluorescence levels between the very lenient and very strict filtering. Given that we expect total fluorescence to scale with cell size, this observation suggests that strict filtering on forward- and side-scatter is not effective for selecting out a subset of cells with similar size.

## Flow cytometer measurements are affected by substantial measurement noise

When using the flow cytometer to estimate single-cell gene expression, we aim to quantify the variation in gene expression across a population of isogenic cells growing in a homogeneous environment. In such conditions, bacteria at different stages of their cell cycle vary by roughly two-fold in size, and their total fluorescence is typically proportional to cell size.

In a previous work we have established that time-lapse microscopy measurements of cells growing in microfluidic devices can measure cell size with an accuracy of around 3% error and GFP copy-number $G$ with an error of about $\sqrt{G}$ [47]. Using such microscopy measurements

on *E. coli* cells carrying a lacZ-GFP fusion gene in its native locus while growing in M9 minimal media with lactose, we find a high correlation between lacZ-GFP levels and cell size (Fig 3, top panels). That is, because lacZ-GFP concentrations fluctuate only moderately from cell to cell, and both size and GFP level measurements have high accuracy, the measured cell length explains around 70% of the variance in total fluorescence.

We calculated the analogous correlation between size and total fluorescence in the flow cytometer for the same strain growing in the same environment, using the scatter signals as representing the cell size. We see that, in contrast to the microscopy measurements, there is only a very weak correlation between total fluorescence and scattering measurements (Fig 3, bottom 4 panels).

The lack of correlation between size and fluorescence measurements in the cytometer strongly suggests that either the fluorescence measurements, the size measurements, or both are much more heavily affected by measurement noise than in the microfluidic experiments. In the following we will look at different sources of noise and how to deal with them.

## Estimating the mean and variance of the fluorescence distribution

As has been observed by others [38], we observed that for virtually all *E. coli* promoters, the distribution of fluorescence levels is fitted very well by a log-normal distribution [23], i.e. the log-fluorescence follows a Gaussian distribution. Our *E-Flow* package fits a Gaussian distribution to the measured log-fluorescence levels of single cells, estimating a mean $\mu$ and variance $v$ for a given population of cells. However, we noticed that, even after filtering events on forward- and side-scatter as explained above, there are still clear outlying events, i.e. with fluorescence levels that lie far outside the range observed for almost all other events. To separate these outliers from valid measurements we modeled the distribution of log-fluorescence levels as a mixture of a Gaussian and a uniform distribution, fitting its parameters using expectation maximization (see Section 3 of S1 File for details). The *E-Flow* package calculates an estimated mean $\mu$ and variance $v$ of the log-fluorescence levels of a set of measurements, together with error bars $\sigma_\mu$ and $\sigma_v$ on these estimates. In addition, transforming from log-fluorescence back to fluorescence in linear scale, the package also calculates mean and variance of the distribution of fluorescence levels, together with error bars (Section 3 in S1 File).

## Autofluorescence estimation

It is well known that the laser used to excite the GFP can also excite other cellular components of the cell, resulting in an "autofluorescence" signal that also occurs in cells without GFP molecules. In addition, the fluorescence signal may also contain a background fluorescence component coming from sources other than the cell's autofluorescence. In order to estimate GFP levels, we need to correct for these other sources of fluorescence and the *E-Flow* package allows for such correction by using measurements of cells that do not express GFP. Let's call $I_M$ the measured fluorescence intensity, $I_T$ the true intensity (deriving from GFP molecules) and $A$ the component from other sources of fluorescence, which for simplicity we will refer to as autofluorescence. We have the relation

$$I_M = I_T + A. \tag{1}$$

Assuming that the component $A$ fluctuates independently from the true fluorescence $I_T$, we

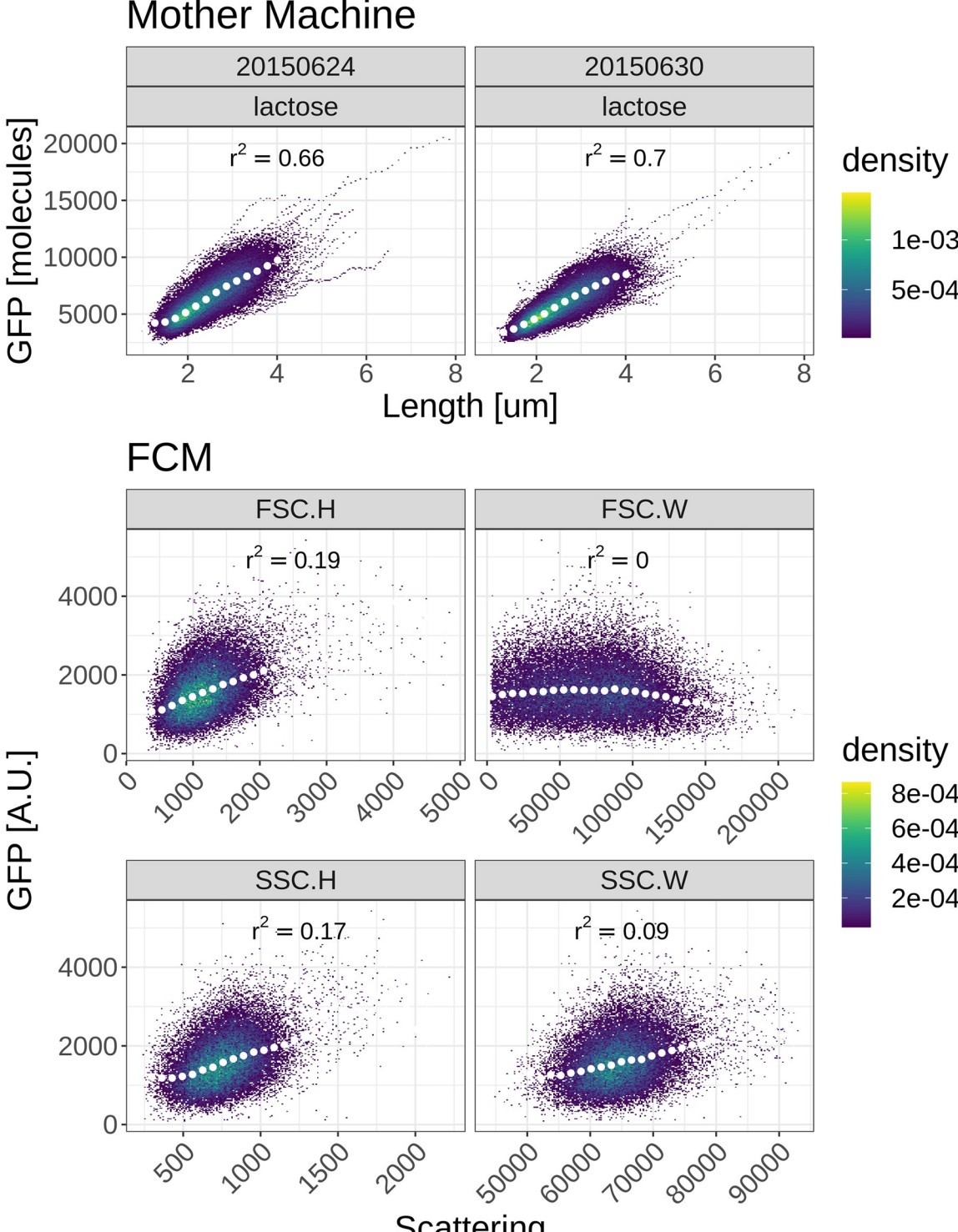

**Fig 3. Correlation between cell size and fluorescence measurements for microscopy and cytometer measurements.** Each panel shows measured GFP fluorescence (vertical axis) and cell size estimates (horizontal axis) of cells growing in M9 minimal media with lactose. The top 2 panels show microscopy measurements from a microfluidic device [47]. The lower 4 panels show fluorescence measurements as a function of size estimates based on forward- (middle 2 panels) or side-scatter (bottom 2 panels) measurements in the flow cytometer (FCM). The squared Pearson correlations between fluorescence and size measurements are indicated in each panel. Note that the color indicates the density of points. The white dots show median values of equally spaced bins along the horizontal axis.

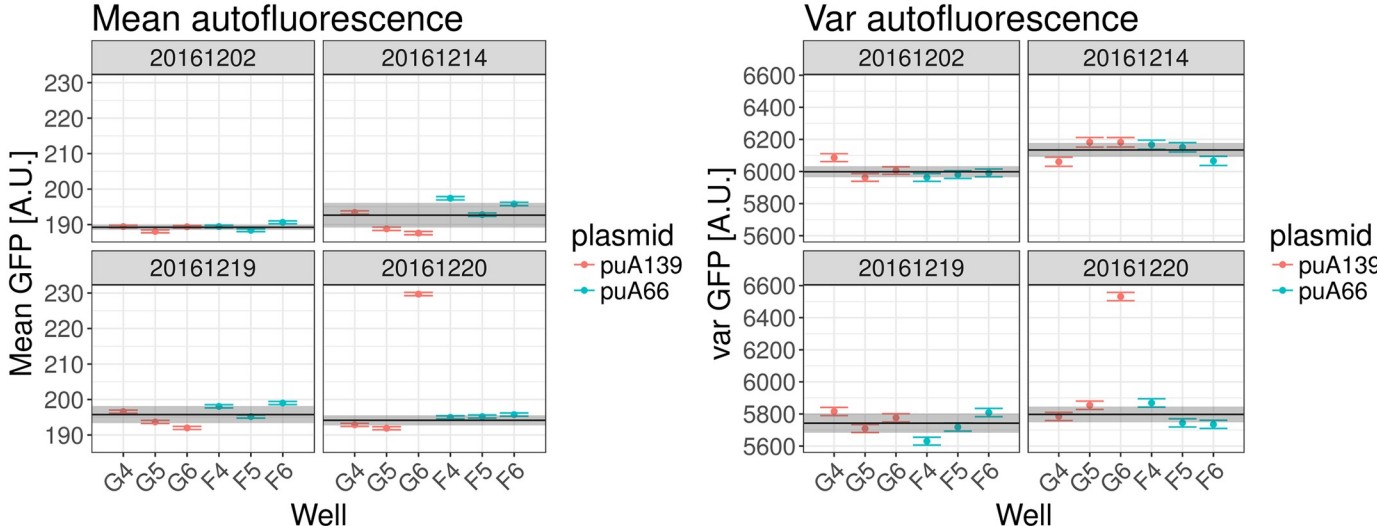

**Fig 4. Autofluorescence measurements.** Each panel shows the measured mean fluorescence (left 4 panels) and variance in fluorescence (right 4 panels) on one day, with each bar indicating the measured value and error bar for one replicate. Two different strains were used (indicated in red and blue) and each was measured in triplicate on each day. The black line and grey bar indicate the estimated averages $\mu_d$ and corresponding error-bars $\sigma_d$ for each day $d$. Note that well G6 on 20/12/2016 appears to be an outlier, possibly due to contamination of the well, which was excluded from the analysis.

obtain

$$\langle I_T \rangle \;=\; \langle I_M \rangle - \langle A \rangle \tag{2a}$$

$$\mathrm{var}(I_T) \;=\; \mathrm{var}(I_M) - \mathrm{var}(A). \tag{2b}$$

Thus, in order to correct for autofluorescence, it suffices to estimate both its mean $\langle A \rangle$ and variance $\mathrm{var}(A)$. These can be easily estimated by performing fluorescence measurements on cells that either lack GFP, or where the GFP gene is known not to be expressed, and applying the same Bayesian mixture model described above. Once $\langle A \rangle$ and $\mathrm{var}(A)$ have been estimated in this way, the true mean and variance of GFP expression in cells carrying an active reporter can be calculated using Eq (2).

We measured autofluorescence levels $A$ using strains carrying two different plasmids not expressing GFP, designed as negative controls (see materials and methods) on 4 different days, measuring each strain in triplicate on each day. Fig 4 shows the estimated mean fluorescences (left 4 panels) and variances in fluorescences (right 4 panels) for each replicate of each strain (red an blue) on each day (one panel per day). Using a procedure described in Section 4 in S1 File, we averaged over different replicates on each day to calculate a mean fluorescence $\mu_d$ for each day (black line in each panel) and an error bar on this estimate (grey region in each panel), and similarly for the variances on each day (right 4 panels). We then additionally averaged over different days to calculate an overall average mean autofluorescence $\bar{\mu}$ and an overall average variance in autofluorescence $\bar{\nu}$ (see Section 4 in S1 File).

### Mean fluorescence levels agree between microscopy and FCM across the entire range of expression levels

Although commercial flow cytometers have been designed to ensure a linear relationship between GFP content and fluorescence measurements over a wide range and previous gene expression studies studies using FCM have operated under this assumption, we here tested this

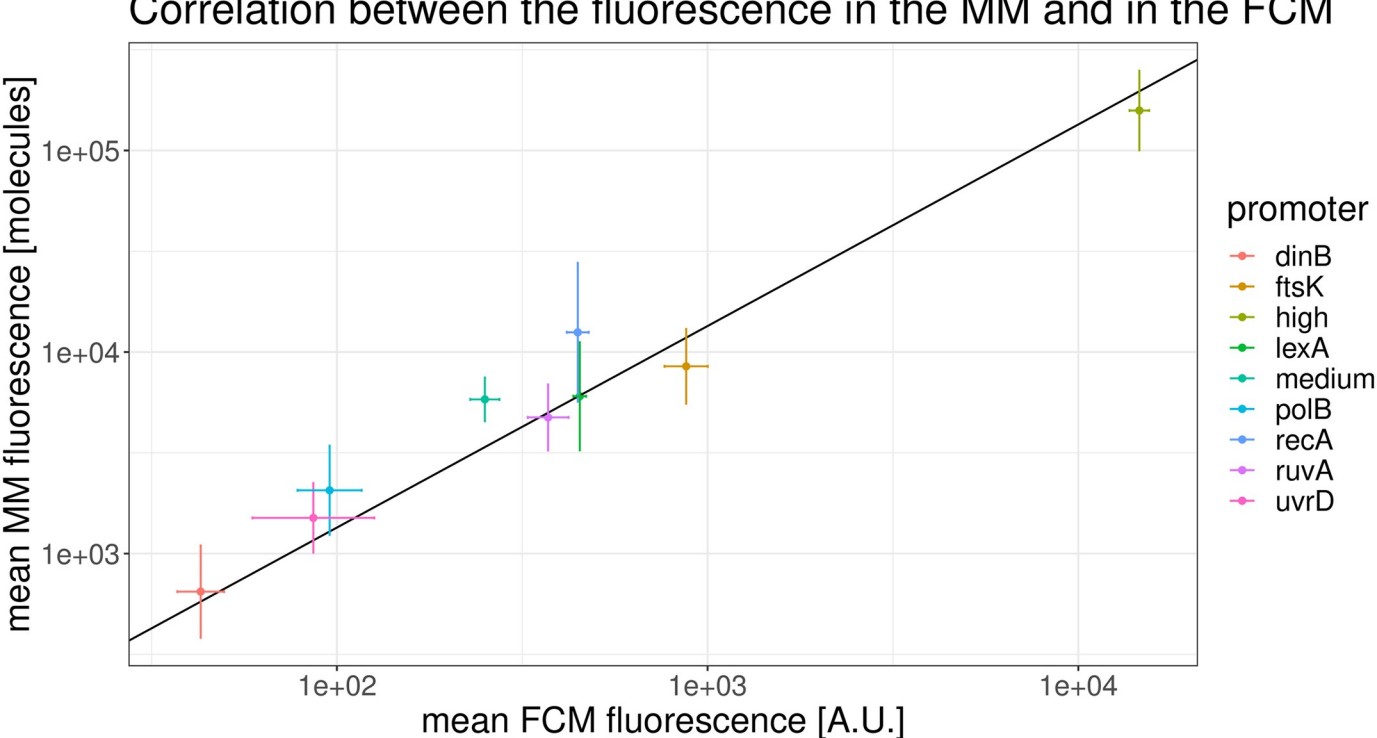

**Fig 5. Estimated mean expression levels of different promoters as estimated by FCM and microscopy.** After correcting for autofluorescence, mean fluorescence levels of different promoters (colors) are perfectly linearly correlated between microscopy and FCM measurements, over the entire range of expression levels. The scales of the axes are in natural log and the error bars show the standard error of the mean. Note that the slope of the black line is 1.

assumption by comparing estimated mean fluorescence levels of different promoters between FCM and microscopy measurements. To do so we calculated the mean fluorescence levels, corrected for autofluorescence, of promoters with a wide range of expression levels using both the FCM and microscopy measurements. As shown in Fig 5, we indeed find that there is a perfectly linear relationship between the average expression levels of the different promoters as estimated by FCM and microscopy, over the entire expression range.

### Cytometer fluorescence measurements exhibit significant shot noise

We used Eq (2) to remove the autofluorescence contribution from the mean expression and variance of the population for a number of different transcriptional reporters and calculated the observed squared coefficient of variation $CV^2$ for each promoter. Next, we took microscopy measurements from our microfluidic setup of the same *E. coli* strains growing in the same conditions and measured $CV^2$ for each of these promoters as well. As shown in the top panel of Fig 6, we observe systematically higher $CV^2$ in the FCM than in the microscopy setup and the difference in the two $CV^2$s decreases almost exactly inversely with the mean expression level.

Since the growth conditions in the FCM and the microfluidic setup were kept as close as possible, the true $CV^2$ of the distribution of total GFP levels should be highly similar, so that the difference between the measured $CV^2$ must derive from measurement noise. Indeed, one source of noise whose contribution to $CV^2$ is expected to scale inversely with mean intensity is shot noise from the photomultiplier tube, whose $CV^2$ scales as 1/mean [52]. Due to this noise, one generally has the following relationship between the measured fluorescence intensity $I_M$

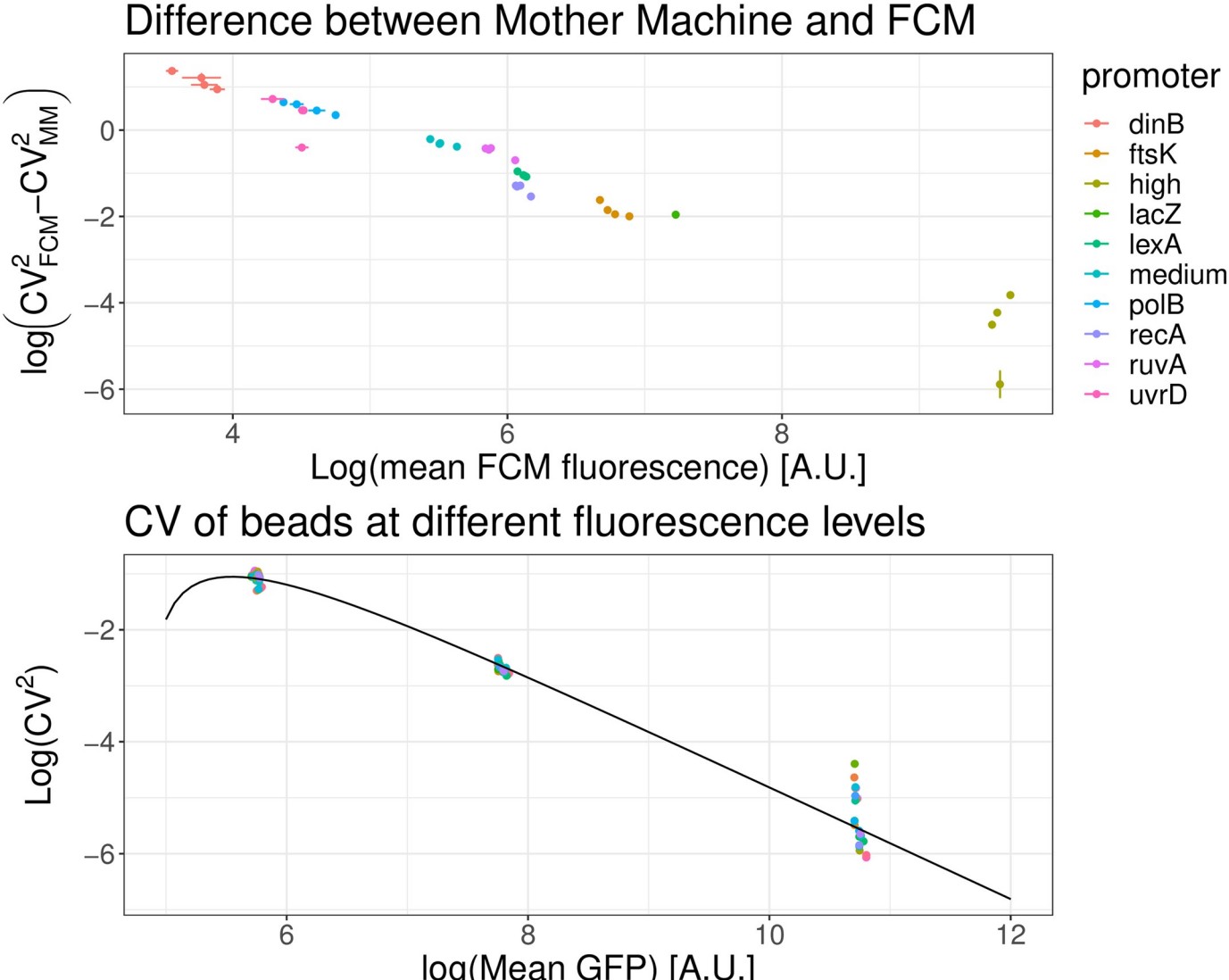

**Fig 6. Difference in $CV^2$ between the FCM and microscopy measurements shows FCM measurements contain substantial shot noise.** *Top*: Difference between the $CV^2$ as measured by the FCM and the microscopy setup for different transcriptional reporters of *E. coli* promoters (colored points). Both axes are shown on a logarithmic scale. The difference in $CV^2$ scales inversely with mean expression. *Bottom*: The observed $CV^2$ of calibration beads of three different intensities also decreases as the inverse of mean intensity and this dependence can be well modeled by shot noise (black line), as given by Eq (3).

and the true intensity $I_T$:

$$I_M = I_T + \epsilon\sqrt{I_T} + O, \tag{3}$$

where $\epsilon$ is a Gaussian random variable with mean 0 and an (unknown) variance $\delta^2$ which quantifies the size of the shot noise. The constant term $O$ is an offset that is added in BD devices in order to prevent the clipping of negative values during the digital conversion, when true intensities $I_T$ are close to zero [51].

Flow cytometers are often calibrated using synthetic fluorescent beads of known intensities and such beads can also be used to estimate the size $\delta$ of the measurement shot noise. As shown in the bottom panel of Fig 6 (and S7 Fig in S1 File) the $CV^2$ of the artificial beads also drops inversely with mean expression. If we assume that the true variation of the beads can be

ignored, we get from Eq (3) that the measured $CV^2$ is

$$CV_M^2 = \frac{\delta^2}{\langle I_M \rangle} - \frac{\delta^2 O}{\langle I_M \rangle^2} \qquad (4)$$

If we define $Y = CV_M^2 \langle I_M \rangle$ and $X = \frac{1}{\langle I_M \rangle}$, we obtain

$$Y = \delta^2 - \delta^2 O X \qquad (5)$$

and we can infer both the strength $\delta$ and the offset $O$ by fitting $Y$ as a simple linear function of $X$. This simple approach leads to an inferred value of $\delta = 13.4$ and $O = 128$. In the Section 5 in S1 File we also present a more sophisticated Bayesian mixture model approach to inferring these quantities, which does not ignore the true variability of the beads, but assumes that the $CV^2$ of the true intensities $I_T$ is the same for all three types of beads. Using this more rigorous procedure, the resulting strength and offset are: $\delta = 12.7 \pm 0.6$, $O = 97 \pm 29$ (S7 Fig in S1 File), which are close to the values we would have obtained with the more simple linear model of Eq (4). Using this result we can now fit the observed $CV^2$ that we expect to see; the fit describes well the observed data, as shown in the bottom panel of Fig 6 (and in the top left panel of S7 Fig in S1 File).

Finally, Section 6 of the S1 File investigates two more subtle technical points that one might think could affect the direct comparison of FCM measurements and microscopy measurement from growth in the microfluidic device. First, one could argue that the age-distributions of the population of cells in the microfluidic device and in a population that is growing exponentially (i.e. as used in the FCM) are different. That is, since in the microfluidic device some newborn daughters are constantly washed out of the growth channels, there are relatively fewer cells close to birth and more cells close to division in the microfluidic device than in a population undergoing exponential growth in bulk (S8 Fig in S1 File). Since total fluorescence correlates with cell size, which again correlates well with time since birth, the access of 'old' cells could in principle effect the distribution of total fluorescence one observes. However, as shown in Section 6.1 in S1 File, we derive theoretically that the effects of the altered age-distribution are small enough to be neglected (S9 Fig in S1 File). Second, since in the microfluidic setup we measure the fluorescence of a cell multiple times during its cell cycle, there are clearly substantial correlations between different measurements and one might wonder whether this could significantly affect the observed statistics. In Section 6.2 in S1 File we show that this effect is also negligible (S9 Fig in S1 File).

## Correcting for autofluorescence and shot noise

After having estimated the mean and variance of the autofluorescence, and the strength of the FCM's shot noise, we can now correct the measured means and variances of transcriptional reporters for these two components. Combining the autofluorescence contribution from Eq (1) and the shot noise component from Eq (3), we can write the measured intensity $I_M$ as

$$I_M = I_T + A_T + \epsilon \sqrt{I_T + A_T} + O, \qquad (6)$$

and the measured autofluorescence as

$$A_M = A_T + \epsilon \sqrt{A_T} + O, \qquad (7)$$

where variables with subscript $T$ correspond to true values and variables with subscript $M$ correspond to measured values, $\epsilon$ is again a Gaussian distributed variable with mean zero and variance $\delta^2$ and $O$ is a constant offset. From these equations we find for the mean and variance of

the measured intensities $I_M$:

$$\langle I_T \rangle = \langle I_M \rangle - \langle A_M \rangle, \tag{8a}$$

and

$$\text{var}(I_T) = \text{var}(I_M) - \text{var}(A_M) - \delta^2 \langle I_T \rangle. \tag{8b}$$

Using these expressions we calculated $\langle I_T \rangle$, $\text{var}(I_T)$ and the resulting $CV^2$ for a set of different *E. coli* promoters and compared the results with the $CV^2$ measured for the same promoters in the microscopy setup. As shown in Fig 7, the estimated $CV^2$ are much closer to the results obtained with the microscopy measurements and the difference no longer systematically depends on the mean expression level. In addition, whereas the $CV^2$ of the raw FCM measurements show little correlation with the $CV^2$ of the microscopy measurements, after correcting for autofluorescence and shot noise there is a much better agreement between the $CV^2$ as measured by the FCM and microscopy (Fig 8).

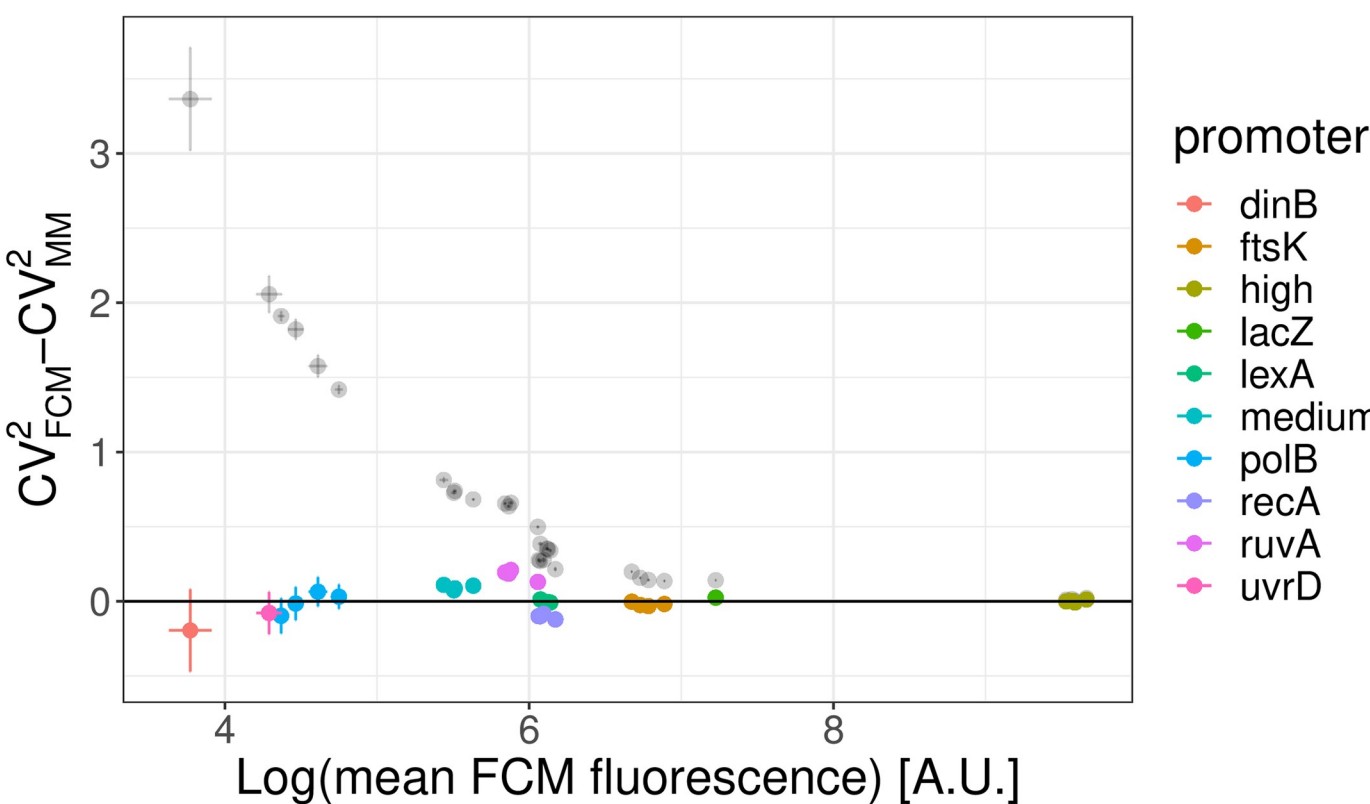

**Fig 7. Comparison of $CV^2$ from FCM and microscope measurements after correcting for autofluorescence and shot noise.** Absolute difference of the $CV^2$ of different transcriptional reporters of native and synthetic *E. coli* promoters as estimated from FCM and microscope measurements. The black transparent dots use uncorrected FCM measurements and reproduce Fig 6 in linear scale, while the colored dots are obtained when using the $CV^2$ that are corrected for the FCM shot noise.

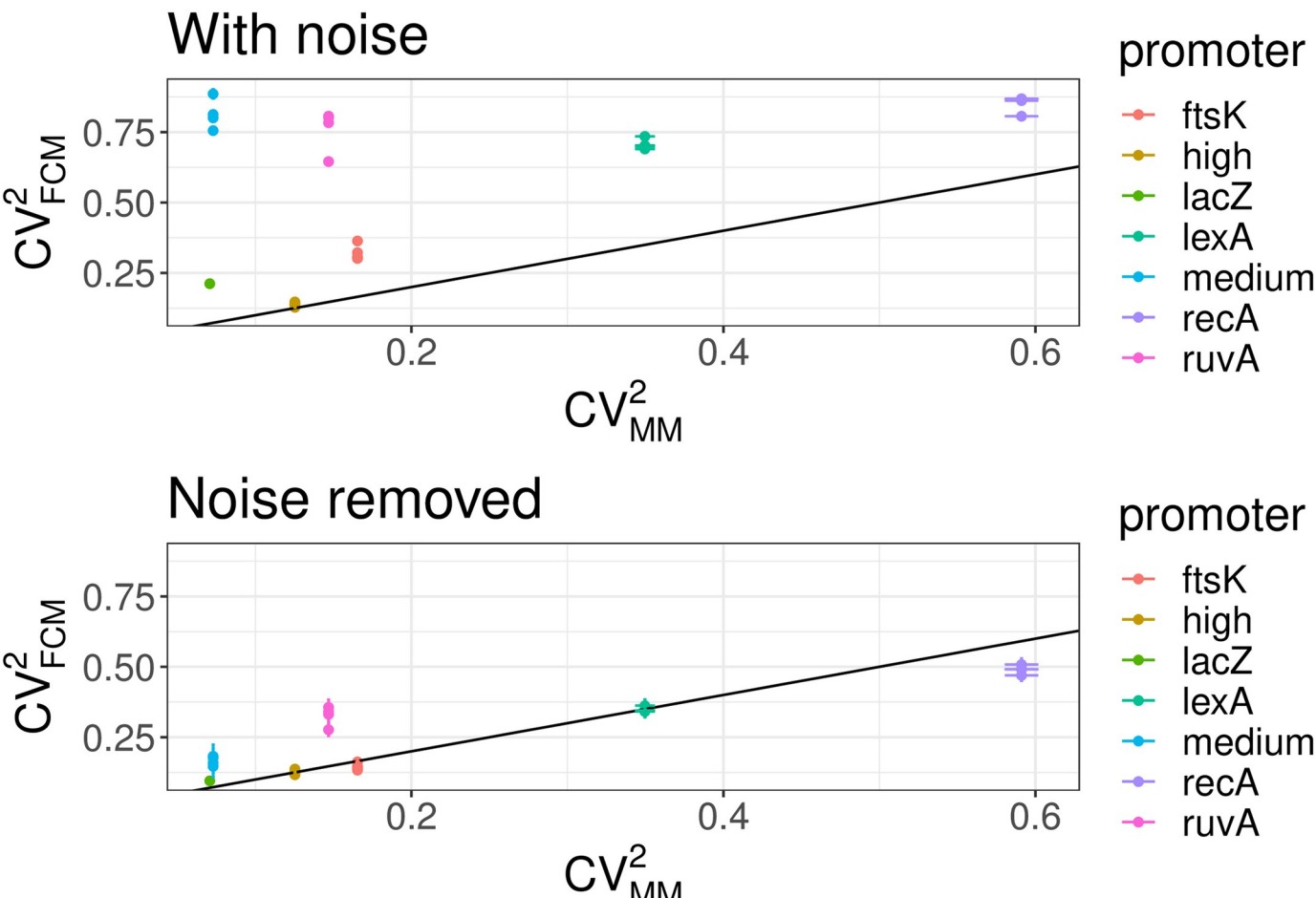

**Fig 8. Correlation of $CV^2$ in the FCM and microscope measurements before and after correcting for autofluorescence and shot noise.** *Top*: The $CV^2$ of the raw FCM fluorescence measurements is consistently higher than the $CV^2$ of fluorescence in the microscope measurements, and there is little correlation between the two. *Bottom*: Once the FCM measurements are corrected for autofluorescence and shot noise, there is now a good agreement between the $CV^2$ as estimated by FCM and microscopy. Measurements for different promoters are indicated by different colors (see legend) and different points of the same color represent replicate FCM measurements. Only promoters expressing more than exp(4) above the background are shown and the black line in both plots is a line with slope 1 and intercept 0.

### Estimating mean and variance of GFP concentration

As shown in Fig 3, microscopy measurements show a strong correlation between the size of the cells and total GFP of the cells, indicating that cell size variations are responsible for a large fraction of the variation in total GFP, and that GFP *concentration* fluctuates significantly less than total GFP. It would thus be desirable to be able to estimate the mean and variance of GFP concentrations from the FCM measurements as well. However, the fact there is a much weaker correlation between raw fluorescence and scatter measurements in FCM (Fig 3) suggests that it may be difficult to accurately estimate GFP concentrations for single cells. In particular, to estimate the GFP concentration of a single cell, we need to not only take the autofluorescence and shot noise of the fluorescence measurement into account, we also need to quantify how the cell's volume relates to the forward- and side-scatter measurement, which is known to be very challenging.

**Scattering signals are non-linear functions of cell size.** The extent to which forward- and side-scatter measurements of FCMs can be used to estimate the size of the measured

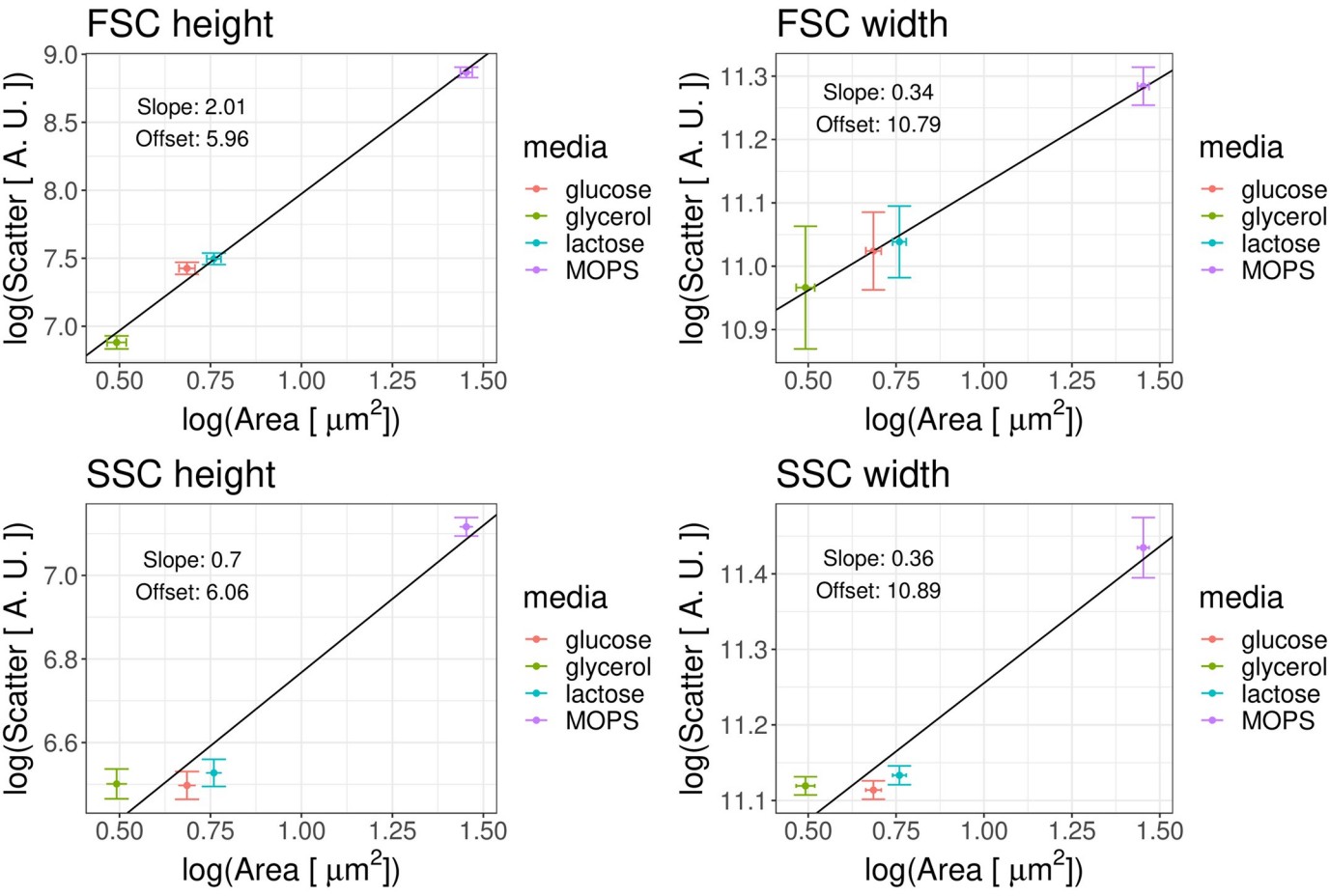

**Fig 9. Average forward- and side-scatter of cells show approximate power-law dependence on average cell size.** Each panels shows the average of the logarithm of one of the four scattering signals, i.e. height or width of either forward- (FSC) or side-scatter (SSC), as a function of the average logarithm of cell area for *E. coli* cells growing in different media (M9 + glucose, glycerol or lactose; MOPS + glucose, see legend) as measured by microscopy [31]. The error bars represent the standard errors of the mean over replicate experiments.

object is a topic of considerable debate in the flow cytometry literature. It is generally assumed that forward scatter mostly reflects cell size, and that side scatter reflects surface properties such as granularity [53]. Several previous studies have established that FCM can be successfully used to distinguish bacteria of different shapes and sizes [40–43], i.e. the average scattering of a population of cells reflects the average size of the cells in the population.

To confirm that, also within our setup, the average size of a population of cells can be inferred from averages of scatter measurements, we made use of flow cytometry measurements from a recent study from our lab in which *E. coli* cells were grown in a number of different conditions and cell sizes were measured using microscopy in each condition [31]. Notably, the growth-rate of the cells varied considerably across these conditions and *E. coli* cells are known to increase size with growth-rate. For each condition, we calculated both the average cell size from the microscopy measurements as well as the average height and width of both forward- and side-scatter.

As shown in Fig 9, we found a very good correlation between forward-scatter and cell size in each condition, confirming results from previous studies that average scatter can indeed be used to estimate average cell size. However, it should be noted that the observed relationship between cell size and scatter is highly non-linear. That is, whereas the height of the forward-

scatter grows approximately quadratically with cell area, the width of the forward-scatter grows approximately as area to the power 1/3. Previous studies indicate that the mathematical relationship between cell size and scattering signal can be highly dependent on the specific experimental setup and is often at odds with predictions of mathematical theories of light scattering [8, 36, 37]. In [54] it is further shown that even if a particular non-linear relation between scattering and single-cell size can be established in a given setting, this relationship is not universal and it can vary even for bacteria of similar sizes and geometric properties. Thus, although we could here make use of the microscopy cell size measurements to calibrate the non-linear relationship between forward-scatter and cell size, it is highly doubtful that this relationship would apply in other settings.

**Scattering signals contain a substantial shot noise component.**   Moreover, in order to be able to estimate GFP concentrations in individual cells, we have to go beyond relating population averages of scatter and size, and estimate sizes of individual cells from the scattering measurements. Several previous studies have reported that it is difficult to use individual scattering measurements to measure variations of the sizes of single cells in a homogeneous population [35, 44–46]. To investigate this within our setup we focused on height of the forward scattering, since based on Fig 9 this signal most strongly correlates with cell size, calculated the $CV^2$ of the scattering as a function of the average scatter, and compared this with the $CV^2$ in cell area as a function of average cell area, as measured by microscope (Fig 10).

We see that, whereas the microscopy measurements indicate that the $CV^2$ in cell size is roughly equal in all conditions, the FCM measurements show a clear decrease of $CV^2$ with mean, similarly to what was observed for the fluorescence signal. As the scatter signal is generated by converting a light signal into an electrical impulse, it is to be expected that scattering measurements are also affected by shot noise, and the results in Fig 10 confirm that this is the case. Thus, in order to estimate the variation in cell sizes from the forward-scatter signals, we not only have to take into account the non-linear relationship between scattering and size, but also the shot noise on the scattering measurements. However, in contrast to the situation with the fluorescence measurements, where we used the calibration beads to estimate the shot noise, we cannot use these beads for estimating the shot noise on the scattering measurements since these are strongly influenced by the geometry and material of the particles. Therefore, the relationship between size and scatter will likely be very different for the beads than for living cells.

In summary, both the complex non-linear relationship between scattering measurements and size, and the absence of a general procedure for estimating the size of the shot noise in the scattering measurements, make it very difficult to estimate the true variability of cell sizes using FCM measurements only. Consequently, we currently do not see a simple way for using FCM measurements to directly measure the GFP concentrations in individual cells.

**FCM measurements can be used to quantify the relative sizes of variation in GFP concentrations of different genes.**   Although we do not believe that, absent of calibration with an independent measurement technology such as microscopy, it is possible to reliably estimate the true sizes of single cells using forward- and side-scattering measurements, FCM measurements can still be used to learn a great deal about the relative noise levels of different genes. Indeed, as confirmed in Fig 7, provided that autofluorescence and shot noise are taken into account, the $CV^2$ of total fluorescence levels of different promoters can be estimated reasonably accurately from FCM fluorescence measurements. Given that each of these fluorescent promoter reporter constructs are embedded in identical cells growing in the same environment, these cells will all exhibit the *same* variation in cell sizes, so that the differences in $CV^2$ in total fluorescence must reflect differences in the $CV^2$ of the GFP concentrations for these reporter constructs.

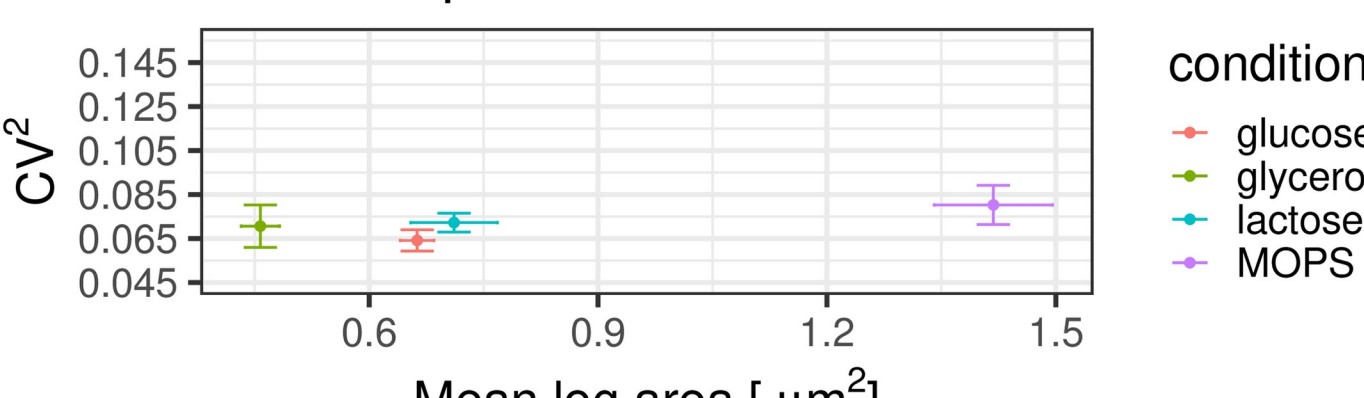

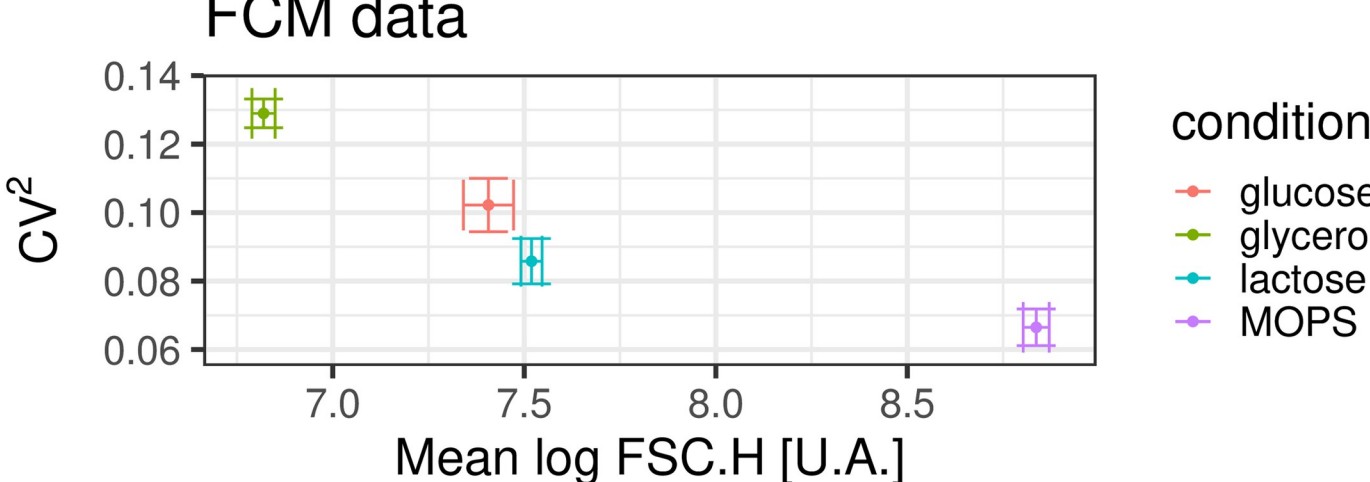

**Fig 10. The $CV^2$ of the scattering distribution is affected by shot noise.** *Top panel*: The $CV^2$ of the cell areas as a function of mean cell area across growth conditions, as measured by microscopy. *Bottom panel*: The $CV^2$ of the height of the forward scattering signal as function of the mean height of forward scattering across growth conditions, as measured by FCM. In both panels the colors corresponds to different growth media as indicated in the legend (M9 + glucose, M9 + glycerol or M9 + lactose, and MOPS + glucose).

Without loss of generality, the total GFP intensity $I$ of a cell can be written as the product $I = C \cdot V$ of GFP concentration $C$ and cell volume $V$, and we can additionally write $C$ as the average concentration $\langle C \rangle$ plus a deviation $\delta_C$, and similarly for volume:

$$I = (\langle C \rangle + \delta_C)(\langle V \rangle + \delta_V), \tag{9}$$

where both $\delta_C$ and $\delta_V$ have average zero.

From the microscopy measurements we know that the fluctuations in the GFP concentration $C$ are approximately independent of fluctuations in cell volume $V$ (S10 Fig in S1 File). Using this, we can derive relationships between both the means and coefficients of variation of

the total GFP $I$, and the concentration $C$ and volume $V$, respectively. We find

$$\langle I \rangle = \langle C \rangle \langle V \rangle \tag{10a}$$

$$CV_I^2 = CV_C^2 + CV_V^2 + CV_C^2 CV_V^2. \tag{10b}$$

We can use this to rewrite the coefficient of variation of concentration, in terms of the coefficient of variation of total GFP (which we have shown how to estimate) and the (unknown) coefficient of variation in cell size, i.e.

$$CV_C^2 = \frac{CV_I^2}{1 + CV_V^2} - \frac{CV_V^2}{1 + CV_V^2}. \tag{11}$$

Thus, if the coefficient of variation of cell volume $CV_V^2$ in the growth condition of interest can be estimated using independent measurements, then Eq (11) can be used to estimate the coefficient of variation of concentration in terms of the $CV_I^2$ for total GFP, as given by Eq (8). Importantly, since the $CV_V^2$ is the same for all reporter constructs, such a measurement would only have to be done once.

Lastly, even if the $CV_V^2$ is not known, we note that it will be the same for each of the promoter reporter constructs. Therefore, the difference $dCV_C^2$ of the coefficients of variation in GFP for two promoters is directly proportional to the difference $dCV_I^2$ in coefficient of variation of total GFP, i.e.

$$dCV_C^2 = \frac{dCV_I^2}{1 + CV_V^2}. \tag{12}$$

Although this still depends on the $CV_V^2$, for all conditions we tested we found that $CV_V^2 \ll 1$, so that a reasonable estimate of the relative size of variation in concentrations is given by simply setting $CV_V^2 = 0$ in the above equation.

## Discussion

Although flow cytometry is an attractive technology for single-cell analysis of gene expression in high-throughput, we have shown that for data from bacterial cells there are a number of challenges to overcome in data analysis in order to obtain accurate quantification. We here developed a number of procedures for measuring single-cell expression distributions in bacteria using FCM data and implemented them in an R package called *E-Flow*.

We first analyzed the forward- and side-scatter signals and their correlation structure. There seems to be little agreement in the literature as to when to use forward-scatter or side-scatter and whether to use height, width or area. We showed that only width and height provide independent measurements and developed a Bayesian mixture model for separating viable cell measurements from debris and other outliers using the full 4-dimensional distribution of forward- and side-scatter measurements. In general the filter we developed is much broader than the very strict gating strategies that are sometimes used and typically only a small fraction of the events are discarded.

We next developed a Bayesian mixture model to estimate the mean and variance in single-cell fluorescences of a population of cells carrying a fluorescent reporter. However, by comparing of the means and variances estimated by FCM with the means and variances estimated from microscopy measurements of the same strains growing in the same conditions, we observed systematic differences because of two effects. First, the amount of autofluorescence per cell differs systematically between FCM and microscopy and we developed methods for

estimating and removing the autofluorescence from the FCM measurements. We show that, after correcting for autofluorescence, there is a perfect agreement between the means of different reporters as estimated by FCM and microscopy, over the entire range of expression levels. However, FCM measurements systematically overestimate the variation in fluorescence levels due to shot noise in the FCM measurement. We developed a method to correct for the contribution of shot noise to the estimated variation that uses calibration beads to estimate the size of the FCM shot noise. We showed that, only after correcting for shot noise do gene expression noise measurements from the FCM converge to those obtained from microscopy measurements. Although the precise size of the shot noise and autofluorescence will likely vary between different flow cytometers, the methods we presented here are general, can be applied to data from any flow cytometer, and provide a step-by-step procedure for both estimating the size of autofluorescence and shot noise, and correcting for these components.

Finally, we investigated whether FCM can be used to directly measure the distribution of GFP concentration across cells by using forward- and side-scatter measurements to estimate the volumes of individual cells. In line with previous work, we show that because scattering measurements depend on cell size in a complex non-linear manner and contain a shot noise component that is difficult to calibrate, it is not possible to accurately estimate the fluctuations in volumes of single cells from scattering measurements. However, because GFP concentration and cell size fluctuate independently across cells, we showed that the relative sizes of GFP fluctuations for different reporter constructs can still be estimated from the variation of total GFP with reasonable accuracy.

## Supporting information

**S1 File. The supplementary materials document provides supplemental methods and supplementary figures.**
(PDF)

## Author Contributions

**Writing – review & editing:** Luca Galbusera, Gwendoline Bellement-Theroue, Arantxa Urchueguia, Thomas Julou, Erik van Nimwegen.

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
