## [Decision Letter · Decision Letter 0]

4 May 2020

PONE-D-20-01335

Using fluorescence flow cytometry data for single-cell gene expression analysis in bacteria

PLOS ONE

Dear Prof. van Nimwegen,

First of all, let me apologize for the huge delay in reviewing your manuscript. The need for qualified reviews and the recent spread of COVID-19 disease deeply impacted on the review time.

After careful reading of past reviewers comments, submission to two additional expert in the field, and my personal evaluation of the manuscript, I feel that it has merit but does not yet fully meet PLOS ONE’s publication criteria as it currently stands. Therefore, we invite you to submit a revised version of the manuscript that addresses the points raised during the review process.

In particular, I agree with reviewer 2 that more extensive reference to previous and closely related work should be presented, to better evidence the clearly innovative aspects of your manuscript.

We would appreciate receiving your revised manuscript by Jun 18 2020 11:59PM. To enhance the reproducibility of your results, we recommend that if applicable you deposit your laboratory protocols in protocols.io, where a protocol can be assigned its own identifier (DOI) such that it can be cited independently in the future. For instructions see: http://journals.plos.org/plosone/s/submission-guidelines#loc-laboratory-protocols

We look forward to receiving your revised manuscript.

Kind regards,

Giovanni Signore

Academic Editor

PLOS ONE

Journal Requirements:

'The funders had no role in study design, data collection and analysis, decision to publish, or preparation of the manuscript.'

Reviewers' comments:

Reviewer's Responses to Questions

**Comments to the Author**

1. Is the manuscript technically sound, and do the data support the conclusions?

Reviewer #1: Yes

Reviewer #2: Partly

2. Has the statistical analysis been performed appropriately and rigorously? 

Reviewer #1: Yes

Reviewer #2: Yes

3. Have the authors made all data underlying the findings in their manuscript fully available?

Reviewer #1: Yes

Reviewer #2: Yes

4. Is the manuscript presented in an intelligible fashion and written in standard English?

Reviewer #1: Yes

Reviewer #2: Yes

5. Review Comments to the Author

Reviewer #1: Galbusera et al, presented a study on the analysis and use of fluorescence flow cytometry data, to study single-cell of bacteria that express a reporter gene at different concentrations. The work is complete and data are sufficient for pubblication. The authors also provide a tool for the measuremnts of atutofluorescence and shot noise of the instrument. However, these results highlight the fact that this technique is not very efficient in the single cell study but rather for the evaluation of the fluorescence signals of an entire bacterial population. The title may mislead the reader who expects to find a single cell analysis approach. I suggest to clarify this point in the title or in the main text.

Reviewer #2: Despite the fact that flow cytometry has been used to quantify gene expression variability in bacteria for almost 20 years now, very little work has been done to systematically analyze the measurement errors in cell size and fluorescence that flow cytometers introduce. It is nice to see that the work presented in this manuscript aims to cover this important gap in our knowledge.

Measuring size and fluorescence of bacteria is no mean feat, as bacterial cells are typically at the limit of detection for flow cytometers. The authors present a plethora of tests and comparisons between flow cytometry and microscopy-based measurements to discover 1) how to interpret flow cytometry signals related to cell size and fluorescence and 2) how to correct them for biases and measurement noise (whenever possible). Below is a list of the main topics addressed by the authors and my comments about them:

1. Choice of signal parameters to monitor: pulse width is typically used to discriminate single particles from doublets. I am therefore not surprised that this parameter is not particularly informative for fluorescence quantification. It is reassuring that the area is proportional to the product of pulse height and width for the calibration beads. Does a similar connection hold for bacterial cells as well?

2. Gating events based on the scatter measurements: in various ways, this is routinely done by several groups that practice flow cytometry, either in a manual or semi-automatic manner. Does the approach proposed by the authors have any significant advantages?

3. Correlation of FSC and SSC with mean cell size: here, the authors are missing an important citation (Volkmer & Heinemann, PLoS ONE, 2011). That work also made a comparison between mean FSC/SSC and cell volume, with very similar conclusions to the ones made here.

4. Use of FSC/SSC to measure individual cell sizes: the authors conclude that FSC/SSC signals do not carry enough information to measure cell size distributions in an E. coli population. Though I would tend to agree with this conclusion, I do not fully agree with the arguments provided. The authors apply equations (1) and (2) both to microscopy and flow cytometry data. While it is clear to me that (1) and (2) are perfectly adequate for size measurements based on microscopy, I am not sure that they apply just as easily in the case of flow cytometry. Specifically, eq. (1) could perhaps look like x_m = f(x_t)+epsilon, i.e. the relationship between measured and actual sizes could be nonlinear at worst, or linear (perhaps with an offset) at best. In either case, the variance of x_m for flow cytometry will not obey (2), even in the simplest scenario of linear scaling without offset.

I think that, before attempting to decompose the measured variance of the flow cytometry signals, the authors should first infer the relationship between x_m and x_t in the case of flow cytometry, for example by using calibration beads of different sizes. Then, using variance propagation they could estimate var(x_t) based on var(x_m) and determine how much information the FSC/SSC signals carry about the size of single cells.

Another point regarding the analysis of this section is that one should probably also suspect the noise in GFP measurements as another source of error. As the authors show later on, the GFP signal requires some significant corrections to correctly report the variability of the cell population. On the other hand, it seems to me that the GFP signals used in Fig. 4 were not corrected for shot noise and autofluorescence. It is therefore plausible that the lack of correlation between GFP and FSC/SSC can be partially attributed to the GFP noise.

5. Gating the fluorescence data & 6. Estimating autofluorescence: the authors perform an analysis which is typically carried out in one or another way by practioners of flow cytometry. However, their approach looks quite principled and systematic (e.g. in the averaging of authofluorescence measurements from different dates).

7. Calibration of fluorescence signal to remove shot noise and autofluorescence: here, the authors perform a nice analysis to remove the additional variability introduced by flow cytometry using calibration beads (an approach similar to what I suggested above for the FSC/SCC analysis), and are eventually able to achieve relatively good agreement between microscopy and flow cytometry measurements. I would have preferred to also see the relative, besides the absolute error in the CV’s, to better understand over which range of GFP concentrations the flow cytometry measurements can provide a reliable estimate of the GFP variability.

Overall, I think that the authors should try to clarify a bit more which steps of their calibration pipeline differ or improve upon procedures carried out by other groups. For example, the fitting of mixture models is also doable with commercially available flow cytometry software, as well as with user-generated code. I would recommend that they revisit the analysis of FSC/SSC and individual cell size.

Some further minor points and questions:

- Strains and growth conditions section (M&M) needs some streamlining to better organize the information. Growth media, precultures and dilutions are a bit mixed up for flow cytometry and microscopy experiments, and a bit hard to follow.

- Fig. 3: would it be possible that the authors repeat some of the information of ref. 31 about the medium conditions? For example, what does “salt” mean? How are cells grown in these different media? (is growth balanced, for example?)

- Fig. 4: what are the units of the horizontal axis in the second and third rows? The scattering units used here are different from those of other figures where scatters are displayed.

- Fig. 5: is it correct that the two strains mentioned here are simply the same background strain carrying two different plasmids?

- Fig. S6: I do not understand the x-axis title (mean log (GFP.H)).

6. PLOS authors have the option to publish the peer review history of their article (what does this mean?). If published, this will include your full peer review and any attached files.

Reviewer #1: No

Reviewer #2: No

---

## [Author Response · Author response to Decision Letter 0]

10 Sep 2020

Detailed responses to all comments of the reviewers are in the 'response_to_reviews.pdf' file. A short summary for the editor is also provided in the cover letter.

---

## [Decision Letter · Decision Letter 1]

23 Sep 2020

Using fluorescence flow cytometry data for single-cell gene expression analysis in bacteria

PONE-D-20-01335R1

Dear Dr. van Nimwegen,

We’re pleased to inform you that your manuscript has been judged scientifically suitable for publication and will be formally accepted for publication once it meets all outstanding technical requirements.

Kind regards,

Giovanni Signore

Academic Editor

PLOS ONE

Additional Editor Comments (optional):

Reviewers' comments:

Reviewer's Responses to Questions

**Comments to the Author**

1. If the authors have adequately addressed your comments raised in a previous round of review and you feel that this manuscript is now acceptable for publication, you may indicate that here to bypass the “Comments to the Author” section, enter your conflict of interest statement in the “Confidential to Editor” section, and submit your "Accept" recommendation.

Reviewer #1: All comments have been addressed

Reviewer #2: All comments have been addressed

2. Is the manuscript technically sound, and do the data support the conclusions?

Reviewer #1: Yes

Reviewer #2: Yes

3. Has the statistical analysis been performed appropriately and rigorously? 

Reviewer #1: Yes

Reviewer #2: Yes

4. Have the authors made all data underlying the findings in their manuscript fully available?

Reviewer #1: Yes

Reviewer #2: Yes

5. Is the manuscript presented in an intelligible fashion and written in standard English?

Reviewer #1: Yes

Reviewer #2: Yes

6. Review Comments to the Author

Reviewer #1: The authors addressed all the points raised by reviewers and I think that now the work can be published on PlosOne

Reviewer #2: i would like to thank the authors for addressing all my comments and modifying their manuscript accordingly.

7. PLOS authors have the option to publish the peer review history of their article (what does this mean?). If published, this will include your full peer review and any attached files.

Reviewer #1: No

Reviewer #2: No

---

## [Editor Report · Acceptance letter]

30 Sep 2020

PONE-D-20-01335R1 

Using fluorescence flow cytometry data for single-cell gene expression analysis in bacteria 

Dear Dr. van Nimwegen:

I'm pleased to inform you that your manuscript has been deemed suitable for publication in PLOS ONE. Congratulations! Your manuscript is now with our production department. 

Kind regards, 

on behalf of

Dr. Giovanni Signore 

Academic Editor

PLOS ONE